# Ecology as a New Foundation for Natural Theology

Andreas Gonçalves Lind * and Bruno Nobre *

Faculty of Philosophy and Social Sciences, Catholic University of Portugal, 1649-023 Lisbon, Portugal
* Correspondence: alind@ucp.pt (A.G.L.); bnobresj@ucp.pt (B.N.)

**Abstract:** The erosion of metaphysics that began in Modernity has led to the discredit of the whole project of natural theology as a means to reach God, establish the classical divine attributes, and account for divine action. After the deconstruction of classical metaphysics propelled by thinkers associated with the Protestant tradition and by philosophers affiliated with the Nietzschean critique, it may appear that only an apophatic approach to God would then be possible. However, the attempt to establish a consensual foundation for the theological discourse has not lost its relevance. In this sense, the attempts to revitalize natural theology are most welcome. It would be naive, however, to think that approaches to natural theology based on classical metaphysics will easily gather consensus. This will not happen. The departing point for a renewed and credible approach to natural theology cannot be the theoretical universal reason associated with Modernity, which is no longer acknowledged as a common ground. As such, a viable approach to natural theology has to find a new consensual starting point. The goal of this article is to argue that the emergence of a new ecological urgency and sensibility, which nowadays gather a high degree of consensus, offers an opportunity for the renewal of natural theology. It is our aim: (i) to show the extent to which God grounds the intrinsic value of nature, which, as such, deserves respect, and (ii) to suggest that the reverence for nature may naturally lead contemporary human beings to God.

**Keywords:** natural theology; theism; ecology; ecological sensibility

## 1. Introduction

Since the golden age of ancient philosophy, throughout the Middle Ages, and until the end of Modernity,[1] natural theology was used as a way of lending credibility to the theology of revelation and also as common grounds for both believers of different creeds and unbelievers (Edwards 2013). Since Modernity, classical metaphysics, which had been taken as the starting point of natural theology, has been subjected to successive critiques. As a result, the foundation of natural theology was progressively eroded, to the point that this approach to theology was no longer able to fulfill its classical tasks, namely lending rational credibility to the Christian revelation. Moreover, this approach was also unable to offer a consensual platform that could sustain the dialogue between believers of different creeds and non-believers. Natural theology became confined to some Catholic intellectual circles, especially seminaries. In the last few decades, however, natural theology went through a revival (McGrath 2008), in particular within analytic philosophy (Taliaferro 2009).

In the context of our multicultural world, the role of natural theology becomes more important than ever. In fact, natural theology could function, once again, as a tool for the dialogue between parts with different perspectives concerning God and religion. According to our view, however, natural theology will not be able to assume this role, at least not in a satisfactory way, unless a "consensual" starting point will be found. This is the first thesis we propose in this article. The second thesis is that the "ecological sensibility" that has emerged in our times, and which is progressively becoming consensual, may offer such a foundation or starting point. In this sense, it is our aim to sketch a natural theology that assumes ecology as its foundation. The relation between ecology and natural theology has recently been explored by Christopher Southgate (cf. Southgate 2013). In his approach,

Southgate tries to read the ecosystems, using, at the same time, scientific and poetical or contemplative methods, in the light of the logic of "gift" (cf. Southgate 2013, pp. 467–68, 472). Our approach is a different one. We show that the ecological concerns are better founded by a theistic perspective according to which the intrinsic and objective value of nature, perceived by contemporary human beings as an imperative to respect nature, is established by its dependence upon God as its Creator. Our approach to a natural theology from ecology makes use of John Rodman's notion of "ecological sensibility", which is often used by the deep "ecology movement".

The outline of the article is the following: we will begin, in Section 2, with a historical overview of natural theology, aiming to show that, from Modernity on, natural theology was progressively deprived of a consensual starting point, which undermined its credibility and limited its ability to fulfill the functions usually ascribed to it. In the following section, a natural theology founded on ecology will be outlined.

## 2. The Decline and Resurgence of Natural Theology

Natural theology may be defined as "the practice of philosophically reflecting on the existence and nature of God independent of real or apparent divine revelation or scripture" (Taliaferro 2009, p. 1).[2] It is not difficult to identify different reasons why one may want to engage in the project of natural theology. On the one hand, natural theology is frequently understood as a preliminary on which theology of revelation may be founded. Understood in this way, natural theology assumes a "foundational role" (Ferguson 2006, p. 384). On the other hand, natural theology may also be regarded as a kind of common ground for believers of different creeds or even for believers and unbelievers, in the sense that no assent to revelation is required. Understood in this way, natural theology becomes an apologetical tool for believers who wish to engage in dialogue with unbelievers or with believers who do not share the same faith (Ferguson 2006, p. 384). Regardless of the role one ascribes to this branch of theology, it becomes clear that, in order to achieve its goal, natural theology has to depart from shared beliefs or presuppositions that remain unquestioned and are thus regarded as consensual and self-evident.

Traditionally, natural theology "is contrasted with *revealed theology*" (Taliaferro 2009, p. 1) to the extent that it seeks to demonstrate the Existence of God and to understand the divine essence through the exclusive use of universal reason, which was considered as being common to both pagans and Christian philosophers. In this sense, the divine Logos was understood as the common ground for both Christian and non-Christian thinkers, and as such, the intelligibility of the universe was presupposed as the firm foundation of natural theology. In this context, the so-called Church Fathers, such as Justin and Augustine, considered themselves as heirs not only of the Old Testament prophets and of the Apostles, but also heirs of the philosophers of Greek antiquity, such as Socrates, Xenophanes, and Plato, who reached a certain notion of God as the supreme Good by reason alone (cf. Ratzinger 2004, pp. 138–39).

In his classic *Introduction to Christianity*, Ratzinger refers, in this regard, to a primordial "option for the Logos". "Early Christianity", Ratzinger affirms, "boldly and resolutely made its choice and carried out its purification by deciding for the God of the philosophers and against the gods of the various religions." (Ratzinger 2004, p. 137). In this way, natural theology becomes possible insofar as there is continuity, without a radical rupture, between the God of the philosophers and the biblical God of Abraham, Isaac, and Jacob, even if the two do not fully coincide. "The Christian faith opted (. . . ) against the gods of the various religions and in favor of the God of the philosophers, that is, against the myth of custom and in favor of the truth of Being itself and nothing else" (Ratzinger 2004, p. 142).

Within this tradition, during the Middle Ages, natural theology, which was not yet explicitly acknowledged as a separate philosophical or theological discipline, made use of classical metaphysics and logic, which by then were regarded as being prior to any worldview. This was due to the fact that medievals conceived the natural world as a reflection of the divine essence (cf. Hall 2013, p. 58). In the sense that the natural world exists as a participation in the divine Logos, there is a metaphysical bridge between the created order

and its Creator, which means that the contemplation and study of nature may lead to God, through the so-called "cosmological ways." Aquinas, in particular, used the most advanced "science" of his time, the Aristotelian synthesis, to conclude by philosophical means alone that God must exist as a prior cause of contingent beings (cf. Mackie 1982, pp. 87–92).

With the dawn of Modernity, classical metaphysics was successively challenged by the most influential philosophers and theologians of the time. Classical metaphysics was, as a matter of fact, progressively abandoned as an outdated and no longer consensual body of knowledge. Different historical explanations for this development have been pointed out: the emergence of the nominalist thought in the late Middle Ages, the Protestant Reformation, the birth of science and its emphasis on empirical knowledge, or even the religious wars that devastated Europe after the Reformation. It is unlikely that only one of these factors, when taken in isolation, can explain the progressive abandonment of classical metaphysics. It is much more likely that the erosion of metaphysics happened as the convergence of all these factors, allied to the reduction of teleology to a mere heuristic function (Ginsborg 2006), and some others that have not been mentioned. What is beyond any doubt is that Hume's skepticism and Kantian transcendentalism created a philosophical horizon that contributed to making the project of natural philosophy inviable (cf. Hartshorne 1989, pp. x–xi). According to John Milbank, this decline of metaphysics began in the High Middle Ages with the nominalism of the Franciscan school, from which would emerge the Protestant Reformation and, later, the Kantian type of agnosticism concerning theoretical reason. In the words of John Milbank,

> Later, in the case of Kant, there is a return to the skeptical horizon opened out by Ockham, and a fulfilment of it in terms of the theoretical bracketing of God which ensures a reigning agnosticism as to our knowledge of 'objects', now firmly confined to the screen of phenomena (. . . ) This extreme skeptical nominalism seems to open out a greater role for the constructive subject. (Milbank 2013, p. 83)

It was imposed, in this way, a Christian tradition different from the one Ratzinger points out when he refers to the "primordial option for the Logos". It is a Christianity with a fideist tendency that denies even the possibility of natural theology. Already in the 20th century, Barth assumes himself to be an heir to this tradition. "Barth's rejection of natural theology is robust and unequivocal. [. . . ] His major concern is that we should see God only as he has graciously revealed himself to us, namely in the person of Jesus Christ. Any other approach is to seek to have God on our own, human terms" (Holder 2013, p. 121).

Despite the fact that Modernity witnessed the progressive abandonment of classical metaphysics, the project of natural theology became all the more invigorated. Its starting point, however, could no longer be the same. The erosion of classical metaphysics was accompanied by a correlative increase in the prestige of the natural sciences, which came to be seen as providing an objective and universal knowledge of nature based upon reason and empirical data. In short, science became the new consensus. As such, it should not come as a surprise that during Modernity, the natural sciences were progressively taken as the starting point of natural theology. As Scott Mandelbrote affirms, "natural theology was able to tame and incorporate the increasingly prevalent idiom of a mechanical philosophy from the mid-seventeenth century onwards and to accommodate the increasing mathematization of nature" (Mandelbrote 2013, p. 76).

Until the publication of Darwin's *Origin of Species*, in 1859, natural theology can be understood as a reaction against Hume's skepticism and the Kantian type of agnosticism. From Darwin on, a new tendency emerged, which consisted in the formulation of new versions of the design argument (cf. Ferguson 2006, pp. 380–81). In fact, before the publication of Darwin's *Origin of Species*, a good example of this kind of natural theology would be William Paley's attempt to build arguments in favor of God's existence using the biological developments of his time. Paley's desire to find a firm foundation for his natural theology is explicitly acknowledged: "In all cases, wherein the mind feels itself in danger of being confounded by variety, it is sure to rest upon a few strong points, or perhaps upon

a single instance." For the author of the classic *Natural Theology*, this "strong point" was the knowledge provided by the natural sciences, in particular, human anatomy:

> For my part, I take my stand in human anatomy: and the examples of mechanism I should be apt to draw out from the copious catalogue which it supplies, are the pivot upon which the head turns, the ligament within the socket of the hip joint, the pulley or trochlear muscle of the eye, the epiglottis, the bandages which tie down the tendons of the wrist and instep, the slit or perforated muscles at the hands and feet, the knitting of the intestines to the mesentery, the course of the chyle into the blood, and the constitution of the sexes as extended throughout the whole of the animal creation. by variety, it is sure to rest upon a few strong points, or perhaps upon a single instance. [...] And, upon these occasions, it is a matter of incalculable use to feel our foundation; to find a support in argument for what we had taken up upon authority. In the present case, the arguments upon which the conclusion rests, are exactly such, as a truth of universal concern ought to rest upon. 'They are sufficiently open to the views and capacities of the unlearned, at the same time that they acquire new strength and luster from the discoveries of the learned.' If they had been altogether abstruse and recondite, they would not have found their way to the understandings of the mass of mankind; if they had been merely popular; they might have wanted solidity. (Paley 2006, pp. 277–78)

Beginning in 1859, the Darwinian theory of natural evolution plunged natural theology into a huge crisis, as it found an explanation for the development of life on planet earth without an exogenous element such as divine intelligence. In other words, with the immanent dynamism of natural selection, it is no longer necessary to postulate a creator who created and inculcated an end to the universe and its life forms (cf. Haught 2010, pp. 1–9). Even an author like Michael Ruse, for whom it is possible to be both a Darwinist and a Christian (cf. Ruse 2001, p. 217), recognizes the difficulties that the theory of evolution raises for natural theology. In fact, numerous authors, such as Richard Dawkins (cf. Dawkins 2006) and Daniel Dennett (cf. Dennett 2007), seek to base their atheistic positions on evolution. We should not forget, in this regard, the reformulations of the cosmological argument of design that developed in the 20th century in contact with the theory of evolution from the very beginning. We refer to authors such as Pierre Teilhard de Chardin (Teilhard de Chardin 2002) and, more recently, Alistair McGrath (McGrath 2007) in his refutation of Richard Dawkins' new atheism. It is clear, however, that a consensus has broken down regarding the logical necessity of postulating God as the origin of the order of the universe and life on the planet. It is important to remark, at this stage, that, besides Darwinism, also the scientific developments that took place during the 19th and 20th centuries, especially within the realm of physics and cosmology, contributed to the discredit of natural theology. In fact, for authors like Stephen Hawking or Steven Weinberg, Big Bang cosmology renders unnecessary to postulate the existence of God in order to explain the origin and development of the universe (cf. Dowe 2005, pp. 142–69).

Later on, with the emergence of contemporary philosophy, metaphysics came under attack from two different fronts. On one side, logical positivism declared every proposition that is not analytic and which cannot be empirically verified as deprived of meaning. This is the content of the principle of verification. Because it does not fulfill the requirements of the verification principle, metaphysics, and, as a consequence, the theological discourse, including natural theology, were rendered meaningless, and as such, placed outside the range of scientific reason. Logical positivism, however, became "prey" to its own requirements, and although it went through a series of reformulations, it was eventually declared as inviable, which opened the way for the resurgence of natural theology. Indeed, particularly since the 1970s, Vienna Circle's criterion of verificationism has fallen into discredit. Thus, new formulations of certain classical arguments have emerged within the realm of analytic philosophy (cf. Kwan 2009, p. 502), especially engendered within modal logic. As W. L. Craig and J. P. Moreland affirm,

The collapse of positivism and its attendant verification principle of meaning was undoubtedly the most important philosophical event of the twentieth century. Their demise heralded a resurgence of metaphysics, along with other traditional problems of philosophy that verificationism had suppressed. Accompanying this resurgence has come something new and altogether unanticipated: a renaissance in Christian philosophy. (Craig and Moreland 2009, p. ix)

It is important to notice that this resurgence of natural theology became especially vigorous within analytic philosophy. Within this framework, a whole series of new retrievals of the classical arguments for the Existence of God have been proposed. It is in this context that Quentin Smith asserted that "God is not 'dead' in academia; he returned to life in the late 1960s and is now alive and well in his last academic stronghold, philosophy departments." The author goes to the point of suggesting what he calls a "desecularization of academia" (Smith 2001, p. 4). To be sure, this "desecularization" should not be understood as if natural theology received a generalized assent. It means, rather, that philosophical reflection about God tends to be considered legitimate, even if there is no consensus about the possibility of affirming or denying God by means of philosophical argumentation.

On the other side, authors such as Marx, Nietzsche, Freud, and Heidegger have also made crucial contributions that moved philosophy away from natural theology. While the three "masters of suspicion" destroyed the widespread consensus that we hold, as cognitive subjects, a neutral and universal reason (cf. Westphal 1993, pp. 14–15), the Heideggerian philosophy of Being established an insurmountable "ontological difference" between the *Sein selbst* and the human *Dasein*. While "to Nietzsche, the senses are not the deceivers, but reason is" (Hovey 2008, p. 59), "Heidegger takes the created–creator dichotomy to reflect a particular conception of being" in order to show that "both God and creatures are distinct beings that participate in being as the commonality of having an ultimate explanation" that is not known (Dillard 2016, p. 15). Therefore, as Allan Megill states referring to Nietzsche and Heidegger, whom he calls "prophets of extremity", these authors have put both "the 'God of the philosophers' and of the 'God of the Bible'" in crisis (cf. Megill 1987, p. xii).

In this way, one may say that with the dawn of contemporary philosophy, the challenge to metaphysical reasoning, which came from different fronts, was raised to a new level. The main "option for the Logos" was deconstructed. Two reasons may be identified: (i) following the Nietzschean narrative, one may argue that the metaphysical concepts and syllogistic logic operative in natural theology have their origin not in the universal Logos but rather in unconscious desires; (ii) within Heideggerian philosophy, the resemblance between the Creator and God's creatures is radically dissolved in the abyss associated with the so-called "ontological difference".

What becomes clear from this discussion is that: (i) natural theology needs a consensual foundation, which must be a set of beliefs and presuppositions that are not seriously questioned by the majority; this consensual basis is different in different philosophical frameworks; (ii) the different critical stance of contemporary philosophy undermined all the classical foundations for natural theology; the Greek Logos, the universal and neutral reason of Modernity, and even science have lost its "consensual" character, which renders difficult, if not impossible, the task of building a natural theology. Thus, in order for natural theology to have a future, it must find a new generalized consensus distinct from theoretical, neutral, and universal rationality. Which "consensual" starting point, within the contemporary worldview, is worth using? Our main thesis consists in identifying this "consensual" basis for natural theology with the ecological sensibility of contemporary men and women.

## 3. Natural Theology from Ecological Sensibility

During the last decades, an acute awareness emerged about the ecological problems that affect our world, which are being caused, at least in part, by human action (cf. Liverman 2015, p. 304). The list of ecological problems that humanity is now facing is dramatic. The waste and throwaway culture has produced, in many different places

of our world, high levels of pollution, both atmospheric and that caused by solid and liquid residues. According to the majority of climate scientists, the massive emission of greenhouse gases associated with human activity is responsible for global warming and important changes in the climate. The problem is worsened by the accelerated deforestation of some regions of the planet. Climate change has dramatic implications at the environmental, social, economic, and political levels. In many regions, natural resources are being depleted, and potable water becomes a scarce resource. The pressure of human activity on the natural world is leading to the loss of biodiversity. The repercussions on the quality of human life are becoming noticeable. It is no longer possible to deny that humanity is facing a problem with dramatic proportions that should be dealt with resolutely and as soon as possible. Despite the fact that some fringes of the population are still in denial, a general consensus is emerging about the need to face the so-called ecological crisis. Political interests still resist the implementation of more robust measures in order to face the different dimensions of the environmental problem created by human activity, which should not be understood as a lack of consensus, especially among the scientific community (cf. Zimmerer 2015, p. 154).[3]

It is important to note that the emerging consensus about ecology gathers people from different ideological, political, and religious perspectives. Indeed, authors as different as Pope Francis (Francis 2015), the feminist Émile Hache (Hache 2019), or the philosopher of "deep ecology" Arne Næss (Næss 1993) have all expressed their concerns about ecological issues. In the context of climate change, probably accelerated by direct human action, ecological concerns for the preservation of the planet and the respect for nature now seem to be widely shared by all spheres of society, believers of various religions and non-believers alike (cf. Boff 2010, p. 344). As Pope Francis affirms in his social encyclical *Laudato Si'*, "a very solid scientific consensus indicates that we are presently witnessing a disturbing warming of the climatic system" (Francis 2015, sec. 23). In this way, it is unquestionable that a "global consensus" is emerging regarding the need for ecological care (cf. Francis 2015, sec. 164).

It is our thesis that the emerging consensus around ecology may offer a new foundation for a renewed and credible approach to natural theology. To be sure, we do not intend to say that ecology will inevitably lead to the affirmation of God's existence or any of the divine attributes. What we are trying to say is that natural theology needs a hinge acknowledged, at least to a certain extent, as consensual. Ecology, as we have argued, appears to be a new hinge aligned with the contemporary mindset. The task we will now undertake is to show to which extent the ecological sensibility may lead to God.

It is important to clarify, from the onset, the type of natural theology we have in mind. David Ferguson recognizes that one of the functions of contemporary natural theology is to establish an interdisciplinary link between revelation and the natural and human sciences. "A fifth task of natural theology [he says] might be discerned in the perceived need to display the ways in which the essential claims of revelation can coexist in positive relation to the best insights available from other disciples and fields of knowledge" (Ferguson 2006, p. 387). While Ferguson focuses mainly on "history" and the "natural sciences", in our case, this interdisciplinarity concerns more the interplay between theology and ecology.

In order to propose a preliminary approach to a natural theology founded on the ecological consensus we have described, it is crucial to identify the characteristics of contemporary "ecological sensibility", an expression that was coined by John Rodman in the mid-1990s. According to him, the "three major components of an Ecological Sensibility" are: (i) "a theory of value that recognizes intrinsic value in nature"; (ii) "a metaphysics that takes account of the reality and importance of relationships and systems as well as of individuals"; (iii) "an ethics that includes such duties as noninterference with natural processes, resistance to human acts and policies that violate the noninterference principle, limited intervention to repair environmental damage in extreme circumstances, and a style of cohabitation that involves the knowledgeable, respectful, and restrained use of nature" (cf. Rodman 1995, p. 126).

We are well aware that Rodman, as well as many of the authors associated with the "deep ecology" movement, is not committed to any kind of theism. However, we will now show how the possibility of theism arises from the three major components of ecological sensibility mentioned above.

The question at this stage is how to substantiate the "intrinsic value" of a nature that is by essence finite or contingent. By "intrinsic value", it is understood that nature must be respected and preserved regardless of the benefits that human beings can extract from it. According to Rodman's first principle, "one ought not to treat with disrespect or use as a mere means anything that has a telos or end of its own—anything that is autonomous in the basic sense of having a capacity for internal self-direction and self-regulation" (Rodman 1995, p. 126). In our understanding, Rodman lays out this principle more as an axiom than as the conclusion of an argument. We are well aware that it is possible to object that Rodman is simply drawing from Aristotelian metaphysics in order to show that each being has its own *telos*, proper to its essence, to conclude that they have an "intrinsic value" (Rodman 1995, p. 126). As such, insofar as nature has its own finality, independently of the value it may have for humans, each being is endowed with an intrinsic value. It is not only a matter of recognizing, at the theoretical level, that both the cosmos and each being inhabiting it have their own metaphysical value. More to the point, the question that arises is why humans should respect each one of these beings, according to their corresponding nature. In this way, Rodman's first principle could be rephrased as follows: "nature deserves my respect and reverence in practice, i.e., in the concrete of my life." In our reformulation of the principle of the "intrinsic value", we intend to highlight its subjective dimension, or, in other words, the conviction experienced by the human person.

In order to show how the principle of "intrinsic value" is better accommodated within a theistic perspective, we will revisit Clodovis Boff's critique of ecological nihilism and immanentism (Boff 2010). According to Boff, any immanentist perspective, i.e., closed to transcendence, appears to be fragile with regard to its capacity to sustain the conviction that one must respect a finite nature. In fact, if the world is finite and limited, i.e., not eternal; if everything will eventually reach its end; if I will not be rendered accountable for the way I have used nature; and, if I have only one life to live, then what is the point of respecting and reverencing nature? Within such a nihilistic horizon, all that could matter is immediate pleasure or success. As a consequence, hedonism imposes itself as an attempt to respond to the *taedium vitae*, which is not able to be overcome. The lack of ecological sensibility is, therefore, an expression of a crisis of a lack of meaning. To sum up, an immanentist perspective, including "deep ecology", where there is no place for God, is not capable of making sense of the subjective conviction concerning the preservation of nature in practical life (cf. Boff 2010, pp. 344–45).

It is possible to argue, then, that the notion of "creaturehood", meaning the condition of being a creature, i.e., the state of dependence in relation to the Creator, may bridge this lack of meaning. While Rodman's immanentist perspective grounds the intrinsic value of nature in the autonomy of nature in relation to humankind, within the theistic perspective we are proposing, the value of nature is grounded by the dependence of both nature and the human being on a transcendent Creator. The dependence of nature upon God as its creator may appear, at first sight, to downplay its intrinsic value. This is not necessarily the case, however. In fact, while the value of an autonomous nature is exhausted in the finiteness proper to the space-time dimension, the value of a nature metaphysically dependent on God goes far beyond this dimension, where everything tends to its nadification. In this sense, God appears, not only as a condition of the possibility of an authentic intrinsic value of nature (in objective terms) but also as a key aspect of a narrative, which may sustain the conviction (i.e., the subjective dimension of value) that nature is endowed with an intrinsic value and worth respecting.

Today, new currents are emerging that promote respect for nature, but from a purely immanentist and biocentric perspective. In this sense, it is worth noting that neither of the two perspectives is, in a strict sense, anthropocentric: while Rodman's approach

is "biocentric", the theistic approach is "theocentric". This line of argumentation is not intended to unequivocally demonstrate God's existence. It is more like a "monstration" of God by means of a narrative that nourishes the conviction that *natura* has an intrinsic value and, as such, should be respected and revered.

This approach to natural theology has to face the challenge according to which the ecological problems humanity is currently facing have their roots in the Judeo-Christian tradition. Lynn White was one of the first authors formulating this critique, which he based on a certain hermeneutic of Gen 1:27.28:

> So God created mankind in his own image, in the image of God he created them; male and female he created them. God blessed them and said to them, "Be fruitful and increase in number; fill the earth and subdue it. Rule over the fish in the sea and the birds in the sky and over every living creature that moves on the ground".

According to White's interpretation of these two biblical verses, nature is no more than an object at the disposal of humankind. In White's own words, "Man named all the animals, thus establishing his dominance over them. God planned all of this explicitly for man's benefit and rule: no item in the physical creation had any purpose save to serve man's purposes. And although man's body is made of clay, he is not simply part of nature: he is made in God's image" (White 1967, p. 1205). Needless to say, this interpretation deprives nature of its intrinsic value in the sense meant by Rodman. Or to be more precise, the value of nature becomes totally dependent on its utility for human beings. Understood in this way, the human being becomes a despotic ruler who subjugates nature by means of technology. This is precisely Heidegger's critique of the Judeo-Christian theological tradition. In fact, Heidegger describes how this tradition reduced God to a cause, more specifically to a *causa efficiens* (Heidegger 1977, p. 26). Insofar as human beings are created in the image of God, they too are understood as beings whose main characteristic is the ability to know, predict, control, and transform nature.

However, this particular hermeneutic of Gen 1:27–28 is not inevitable. In fact, during the last decades, alternative interpretations have been proposed, according to which the human being is not a domineer but a steward of nature. In *Laudato si'*, Pope Francis offers a clear synthesis of this alternative reading:

> We are not God. The earth was here before us and it has been given to us. This allows us to respond to the charge that Judeo-Christian thinking, on the basis of the Genesis account which grants man "dominion" over the earth (cf. Gen 1:28), has encouraged the unbridled exploitation of nature by painting him as domineering and destructive by nature. This is not a correct interpretation of the Bible (. . .) The biblical texts are to be read in their context, with an appropriate hermeneutic, recognizing that they tell us to "till and keep" the garden of the world (cf. Gen 2:15). "Tilling" refers to cultivating, ploughing or working, while "keeping" means caring, protecting, overseeing and preserving. This implies a relationship of mutual responsibility between human beings and nature. Each community can take from the bounty of the earth whatever it needs for subsistence, but it also has the duty to protect the earth and to ensure its fruitfulness for coming generations. (Francis 2015, sec. 67)

On the one hand, Gen 1:27–28 should be nuanced with the image, in Gen 2:15, of the human being as gardener and steward of the world. In doing so, the biblical text makes it clear that only God is the Lord of the world and of humankind. Together they are God's creatures (cf. Boff 2010, p. 348). On the other hand, in the biblical text, the use of the verbs "to subdue" and "to rule" are usually applied to the wise king, who takes care of the well-being of every creature entrusted to him (cf. Boff 2010, p. 347).

In this regard, Joseph Ratzinger stresses that Christianity demythologizes the world in the sense that it conceives the world as ordered and intelligible. However, the capacity that the human being has to understand the world should not imply that it can be

used in a merely instrumental way. Ultimately, it is a matter of adhering to the rhythm of nature and to the logic of creation in a movement that is as active as it is passive (cf. Ratzinger 1995, pp. 33–39).

As a final remark, it is worth pointing out that, according to the biblical narrative, it is God who endows nature with its "intrinsic value" when declaring it as being good, in the refrain that repeated throughout the first account of creation: "And God saw that it was good" (Gen 1:9). Thus, Scripture, as a witness to the Judeo-Christian Revelation, affirms the intrinsic goodness of the creation. Of course, this intrinsic goodness nuances the human being's dominion over creation (affirmed in Gen 1) in the sense that nature deserves respect and reverence. This reverence for nature is not exclusive to the biblical tradition and can therefore become a common point among believers of various religions and non-believers.

The biblical notion of "stewardship" underlines the responsibility of the human being in relation to the natural world. Human beings are not only able to acknowledge the objective and intrinsic value of nature but also be responsible for its care and protection. This notion of responsibility, linked with the third component of the "ecological sensibility" described by Rodman, could constitute an alternative foundation for an "ecological" natural theology. If the human being experiences within him or herself the imperative to care for finite nature, one may wonder about the authorship of such a commandment. In other words, if humans understand themselves as stewards and not owners or rulers, one may naturally wonder who the true owner is.

This sketch of the natural theology founded upon ecology is in line with two of the classical ways to reach God by means of natural reason alone, namely the cosmological and the moral arguments in favor of God's existence. To a certain extent, these two ways are integrated into a single argument that departs from the characteristics of the ecological sensibility as a possible new consensual basis for natural theology. In fact, a theist may share this new consensual basis with contemporary people, believers of different religions or Christian denominations and non-believers alike. Moreover, by emphasizing the need to elaborate a narrative capable of nourishing the subjective conviction for the care of nature, our approach takes also into account the deconstruction of classical metaphysics.

## 4. Conclusions

We began this article by revisiting, albeit briefly, the history of natural theology in its fundamental features. In doing so, we have sought to show how natural theology has always been possible on the basis of a broad consensus among Christians, believers of the different religions, agnostics, or atheists. Throughout this millennial history, and as a general rule, universal and neutral reason constituted that consensus, whether when applied to all spheres of knowledge (both in antiquity and during the medieval period) or when restricted to the method of the empirical sciences (during modernity). This consensus was, to a certain extent, broken in the last two centuries, especially with the emergence of positivism and, even more radically, with the postmodern deconstruction of classical metaphysics.

In this sense, if natural theology is to have a future, a new broad consensus must be found within the contemporary worldview. The "ecological sensibility" in its major components may constitute the new starting point that the theist philosopher needs in order to develop new arguments, or new reformulation of the old ones, for God's existence. This approach consists in showing how difficult it is, without theistic presuppositions, to ensure that nature has a value that is intrinsic to it, although dependent upon God's act of creation. In the end, one who might conceive nature closed in its own finite goals will hardly be justified in recognizing, in his or her concrete life, a real value to it. In the long term, the commitment to protect nature, and even the ecological sensibility, may fade away. However, on the contrary, within a theistic horizon, it seems easier to assume, and experience, the justified conviction according to which nature has an intrinsic value, simply because the belief that everything comes from the eternal and loving God nourishes the confidence that nature is essentially good and should be protected as such.

Of course, this is neither an argument nor a demonstration in the most classical sense of the term. It is a bit closer to what Frederick Copleston says to Bertrand Russell in the iconic 1948 debate: "I don't regard religious experience as a strict proof of the existence of God, so the character of the discussion changes somewhat, but I think it's true to say that the best explanation of it is the existence of God" (Russell and Copleston 1957, p. 158). On the one hand, in theoretical terms, when one assumes the creatureliness proper to nature, that is, its provenance from the ultimate God, the objective and intrinsic value of all creatures becomes undeniable. Without God, it seems more difficult to sustain, at this theoretical level, the objective value of a finite nature, condemned to the contingency of the finite world, if not to its nadification. On the other hand, the conviction that nature is intrinsically good because it comes from God who created it in this way fosters in the believer the feeling of moral responsibility in preserving the planet and caring for the beings who inhabit it.

**Author Contributions:** The authors have done all the investigation and all the writing together. No software was used. Both authors have read and agreed to the published version of the manuscript.

**Funding:** This research was funded by the Portuguese Foundation for Science and Technology, grant number UIBD/00683/2020 (Center for Philosophical and Humanistic Studies).

**Institutional Review Board Statement:** Not applicable.

**Informed Consent Statement:** Not applicable.

**Data Availability Statement:** Not applicable.

**Acknowledgments:** We thank Michael Rossmann for the careful revision of the manuscript.

**Conflicts of Interest:** The authors declare no conflict of interest.

## Notes

1. The term "Modernity" has different meanings throughout the literature. Even when it refers to a period in the History of Philosophy, it is difficult to find a consensus as to its beginning and end. In this article, "Modernity" is understood as the period in the History of Philosophy that begins with Descartes in the 17th century and ends with Hegel in the 19th century. It is a perspective according to which natural reason enables human beings not only to understand but also to predict, dominate, and transform nature according to their own will. This way of understanding "Modernity" is linked with the primacy of natural reason and of the scientific method as a way to understand, predict, and control reality.
2. Here is the classical definition, present in the literature, of natural theology. As we will see later, natural theology has different functions today. Additionally, in this article, we will focus on the search for a broad consensus on a theme that can serve as a basis for the natural theology of the future.
3. A recent survey conducted in the USA shows that, at least for younger generations, addressing the problems associated with climate change should be a top priority to ensure the sustainability of the planet for future generations (cf. Funk 2021).

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
