# Peer review of "Ecology as a New Foundation for Natural Theology"

_religions, doi:10.3390/rel12080660_

Round 1

Reviewer 1 Report

As stated, this is an original and well documented study. It should be published, but its overall merit could  be substantially improved by considering some the following:

  1. The definition of natural theology (NT) given in lines 60-61 is OK as far as it goes, but given the stated thesis, namely, to build an ecologically founded NT with consensus as a starting point, it would seem important to build a clearer connection between Judeo-Christian concepts and ecology. The concept of stewardship is an important aspect of this, but it is introduced (lines 392-399) far too late. This should be moved up earlier in the discussion.
  2. I have no quibble with the sources of the decline of NT--the collapse of classical metaphysics, the rise of empirical science, the influences of Hume and Kant (the latter not so much "transcendentalism" [line 115] but more significantly the Kantian reduction of teleology to a mere heuristic, e.g. see Hannah Ginsborg, "Kant's biological teleology and its philosophical significance," in A Companion to Kant [2006]; and Sidan Lofti, "The 'purposiveness' of life: Kant's critique of natural teleology," The Monist 93 [Jan. 2010]: 123-134), and Karl Barth's fideism--all seem reasonable.
  3. Another source in the decline of NT needs to be mentioned: the impact of Darwinian theory. While the theory can function as a catalyst to stronger ecological concerns, other aspects have--indeed can still have--a very contrary influence on ecology. The notion of conquering nature and of manipulating it to one's own uses has a history in Darwinian thought and has been well outlined by Max Horkheimer (1895-1973) in various publications, esp. his Eclipse of Reason (1947) that still carries analytic and explanatory force. The mere mention of Darwin's Origin of Species (lines 144-153) is not enough. Horkheimer's objections to this aspect of Darwinism is thoroughly secular, but important to consider esp. given the central thesis of the paper.
  4. That same paragraph (line 144-153) from Darwin "new versions of the design argument" emerged citing William Paley as an example, but Paley's Natural Theology  (1802) was well BEFORE Darwin's Origin. This needs to be restated. As it stands, it appear inaccurate and a serious historical blunder.
  5. Lines 185-186 make reference to the "resurgence of natural theology" and then makes mention of analytic philosophy being the "heir of logical positivism." If so, that hardly explains the resurgence of NT since logical positivism's hallmarks are it hard verificationism and reductionist materialism vis-a-vis the influential Vienna Circle from the early 20th century through the 1930s. This needs to be restated or explained.
  6. The paragraph (lines 340-344) is OK but could well degenerate into a kind of New Age nature worship. Food for thought. 
  7. Lines 392-409 give the paper a strong ending, but the stewardship piece should be stated up front (see #1).
  8. While scriptural references are not necessary for a coherent NT, this paper needs to state more clearly how and why the Judeo-Christian perspective gives humans dominion over nature on the one hand and yet expects not possession but caring oversight and respect on the other. This connection is key to the argument; otherwise this could easily degenerate into a vague pantheism or worse, a sort of pagan proposal not far from a Wiccan-like hermeticism, which does not appear to be the intention here.

Author Response

Dear reviewer,

Thanks a lot for your suggestions.

In order to make it clear from the start that we are not going to work only with the more classical definition of Natural Theology, that we quoted, we have placed a Footnote (#2) saying the following: “[1] Here is the classical definition, present in the literature, of natural theology. As we will see later, natural theology has different functions today. And, in this article, we will focus on the search for a broad consensus on a theme that can serve as a basis for the natural theology of the future.”

As for the second remark, although the article is not very focused on the details of the erosion of metaphysics from Kant on, we have added a brief paragraph, with the suggested bibliographic reference, on page 4: “It is much more likely that the erosion of metaphysics happened as the convergence of all these factors, allied to the reduction of teleology to a mere heuristic function (Ginsborg 2006).”

The third observation seems fundamental to us. In this sense, in order to refer to the impact of Darwinism on natural theology, we have added the following paragraph on page 5: “Beginning in 1859, the Darwinian theory of natural evolution plunged natural theology into a huge crisis, as it found an explanation for the development of life on planet earth without an exogenous element such as divine intelligence. In other words, with the immanent dynamism of natural selection it is no longer necessary to postulate a creator who created and inculcated an end to the universe and its life forms (cf. Haught 2010, pp. 1-9). And even an author like Michael Ruse, for whom it is possible to be both a Darwinist and a Christian (cf. Ruse 2001, p. 217), recognizes the difficulties that the theory of evolution raises for natural theology. In fact, numerous authors, such as Richard Dawkins (cf. Dawkins 2006) and Daniel Dennett (cf. Dennett 2007), seek to base their atheistic positions on evolution. We should not forget, in this regard, the reformulations of the cosmological argument of design that developed in the 20th century in contact with the theory of evolution from the very beginning. We refer to authors such as Pierre Teilhard de Chardin (Teilhard de Chardin 2002) and, more recently, Alistair McGrath (McGrath 2007) in his refutation of Richard Dawkins’ new atheism. It is clear, however, that a consensus has broken down regarding the logical necessity of postulating God as the origin of the order of the universe and life on the planet.”

Of course, it was necessary to change a little the paragraphs immediately preceding and following in order to give consistency to the text.

Regarding the fourth observation, we have made it clear that Paley's Natural Theology predates the publication of the Origin of Species and the changes that Darwinism brought about, as shown on page 5.

Regarding the fifth observation, we have added a brief bibliographical reference that corroborates the emergence of new reformulations of classical arguments within analytic philosophy: “Indeed, particularly since the 1970s, Vienna Circle’s criterion of verificationism has fallen into discredit. Thus, new formulations of certain classical arguments have emerged within the realm of analytic philosophy (cf. Kwan 2009, p. 502), especially engendered within modal logic.” Namely, we refer to the work of Kwan, Kai-Man. 2009. The argument from religious experience. In The Blackwell Companion to Natural Theology. Edited by William Lane Craig and J. P. Moreland. Oxford: Blackwell, pp. 498-552. In doing so, we have tried to make explicit the fact that the fall of verificationism has allowed an emergence of a resumption of classical arguments (reformulated in a more modern logic) within analytic philosophy.

As for the sixth observation, we have reworded a whole paragraph on page 9, so as to make clear our distancing from the immanentism of certain currents, such as New Age spiritualism. The paragraph looks like this: “Today, new currents are emerging that promote respect for nature, but from a purely immanentist and biocentric perspective. In this sense, it is worth noting that neither of the two perspectives is, in a strict sense, anthropocentric: while Rodman’s approach is ‘biocentric,’ the theistic approach is ‘theocentric.’ This line of argumentation is not intended to unequivocally demonstrate God’s existence. It is more like a ‘monstration’ of God by means of a narrative which nourishes the conviction that natura has an intrinsic value and, as such, should be respected and revered.”

Regarding the last remark, we have added and modified paragraphs on page 10 to make it more explicit how and why the Judeo-Christian perspective gives humans dominion over nature on the one hand and yet expects not possession but caring oversight and respect on the other. More specifically, we have added to Boff's and Pope Francis' exegesis the reference to an analysis by Joseph Ratzinger in which the German theologian delves into this very question. We added the following:

“In this regard, Joseph Ratzinger stresses that Christianity demythologizes the world, in the sense that it conceives the world as ordered and intelligible. However, the capacity that the human being has to understand the world should not imply that it can be used in a merely instrumental way. Ultimately, it is a matter of adhering to the rhythm of nature and to the logic of creation in a movement that is as active as it is passive (cf. Ratzinger 1995, pp. 33-39).

As a final remark, it is worth pointing out that, according to the biblical narrative, it is God who endows nature with its ‘intrinsic value’ when declaring it as being good, in the refrain that repeated throughout the first account of creation: “And God saw that it was good” (Gen 1:9). Thus Scripture, as a witness to the Judeo-Christian Revelation affirms the intrinsic goodness of the creation. Of course, this intrinsic goodness nuances the human being’s dominion over creation (affirmed in Gn 1), in the sense that nature deserves respect and reverence. This reverence for nature is not exclusive to the biblical tradition, and can therefore become a common point among believers of various religions and non-believers.”

Reviewer 2 Report

While classical formulations of  "natural theology" derived from certain metaphysical assumptions have indeed been discredited over the last several centuries, it is now possible to identify prospects for developing a new consensus for something like natural theology--grounded in contemporary emergence of an environmental ethic and an imperative for earth care.  I take that to be the essence of this article's theocentric argument.  From my standpoint that argument has original merit and merits publication even though it draws in passing on some pretty well-worn appeals to "stewardship," reinterpretation of Genesis, etc. 

     Another strength of the article is its historical analysis of what happened to "natural theology," and how notions of nature's "intrinsic value" as end rather than simply instrumental means (with Kantian resonance, too, I might say) could allow for a recovery of consensus in our own day. The references to Laudato Si' and various philosophic analysts are well handled, too.   

     Beyond the correction of minor editorial glitches (see, for example, lines 31 and 407), I 'd recommend two main points of appropriate revision for this piece. So first, I'd wish the Authors to refine a couple of terminological usages that appear throughout the article.  From the start "Modernity," for example, can be variously referenced in historical time as situated somewhere between the 16th century and the present day (or if we're now in the post-modern era, just when did we get there?) , so it would help for the Authors to indicate more clearly their understanding in this regard.  And from the somewhat inaccurate subtitle of "ecology" at line 230, I think that that biological principle of interdependent life forms shouldn't be linguistically conflated, as it is there, with the Authors' actual emphasis on something more like "environmental  awareness," "or an "environmental ethic," or (another better term that the authors do use at times) an "ecological sensibility."

My second main recommendation is that the Authors would do well to make more of those shifts in scientific understanding which (in my view) did even more to discredit classical "natural theology" than the erosion of metaphysics.  Around line 144ff, the article does mention without elaboration the impact of Darwinism on the notion of "design" set forth by Paley and others.  Fair enough so far. But the latter-day revolutions from the 19th c. in geology, physics, and astrophysics certainly have had crucial consequences for theology as well.  I quite agree with the article's sense that "cosmological and moral arguments" might lead toward a rehabilitation of natural theology in our own day.  But the article offers little concrete sense of how that might be so, since no scientifically informed exponents of these possibilities are referenced. At least some allusion to figures such as Pierre Teilhard de Chardin, John Polkinghorne,  Alistair McGrath, or Arnold Benz would be extremely illuminating in this regard, as would mention of cosmological notions such as "fine tuning" and the "anthropic principle."  There are indeed ways in which recent cosmological discoveries now enable us to perceive new pointers--though not proofs--toward the viability of a theocentric worldview such as that affirmed by the Authors.

Author Response

Dear reviewer,

Thanks a lot for your appreciation of our article. We follow your suggestions closely.

In addition to the corrections of the typos mentioned, and an in-depth review of English by a native speaker, we have added a footnote explaining our understanding of Modernity in the article. In addition, we have changed title 3 with the more precise terminology "ecological sensitivity". 

Regarding the second observation, which is more fundamental in terms of content, we have added two long paragraphs on page 5, saying the following: 

“Beginning in 1859, the Darwinian theory of natural evolution plunged natural theology into a huge crisis, as it found an explanation for the development of life on planet earth without an exogenous element such as divine intelligence. In other words, with the immanent dynamism of natural selection it is no longer necessary to postulate a creator who created and inculcated an end to the universe and its life forms (cf. Haught 2010, pp. 1-9). And even an author like Michael Ruse, for whom it is possible to be both a Darwinist and a Christian (cf. Ruse 2001, p. 217), recognizes the difficulties that the theory of evolution raises for natural theology. In fact, numerous authors, such as Richard Dawkins (cf. Dawkins 2006) and Daniel Dennett (cf. Dennett 2007), seek to base their atheistic positions on evolution. We should not forget, in this regard, the reformulations of the cosmological argument of design that developed in the 20th century in contact with the theory of evolution from the very beginning. We refer to authors such as Pierre Teilhard de Chardin (Teilhard de Chardin 2002) and, more recently, Alistair McGrath (McGrath 2007) in his refutation of Richard Dawkins’ new atheism. It is clear, however, that a consensus has broken down regarding the logical necessity of postulating God as the origin of the order of the universe and life on the planet.

Later on, with the emergence of contemporary philosophy, metaphysics came under attack from two different fronts. On one side, logical positivism declared every proposition that is not analytic and which cannot be empirically verified as deprived of meaning. This is the content of the principle of verification. Because it does not fulfill the requirements of the verification principle, metaphysics, and as a consequence the theological discourse, including natural theology, were rendered meaningless, and as such placed outside the range of scientific reason. Logical positivism, however, became ‘prey’ to its own requirements, and although it went through a series of reformulations, it was eventually declared as inviable, which opened the way for the resurgence of natural theology. Indeed, particularly since the 1970s, Vienna Circle’s criterion of verificationism has fallen into discredit. Thus, new formulations of certain classical arguments have emerged within the realm of analytic philosophy (cf. Kwan 2009, p. 502), especially engendered within modal logic.”

Moreover, we have added the following on page 10:

“In this regard, Joseph Ratzinger stresses that Christianity demythologizes the world, in the sense that it conceives the world as ordered and intelligible. However, the capacity that the human being has to understand the world should not imply that it can be used in a merely instrumental way. Ultimately, it is a matter of adhering to the rhythm of nature and to the logic of creation in a movement that is as active as it is passive (cf. Ratzinger 1995, pp. 33-39).

As a final remark, it is worth pointing out that, according to the biblical narrative, it is God who endows nature with its ‘intrinsic value’ when declaring it as being good, in the refrain that repeated throughout the first account of creation: “And God saw that it was good” (Gen 1:9). Thus Scripture, as a witness to the Judeo-Christian Revelation affirms the intrinsic goodness of the creation. Of course, this intrinsic goodness nuances the human being’s dominion over creation (affirmed in Gn 1), in the sense that nature deserves respect and reverence. This reverence for nature is not exclusive to the biblical tradition, and can therefore become a common point among believers of various religions and non-believers.”

Round 2

Reviewer 1 Report

Your changes seem fine and effectively address all of the original comments in my first review.